# Differential Proteomic Analysis of *Listeria monocytogenes* during High-Pressure Processing

**DOI:** 10.3390/biology11081152

**Published:** 2022-07-31

**Authors:** Yi-An Chen, Guan-Wen Chen, Hao-Hsiang Ku, Tsui-Chin Huang, Hsin-Yi Chang, Cheng-I Wei, Yung-Hsiang Tsai, Tai-Yuan Chen

**Affiliations:** 1Department of Food Science, National Taiwan Ocean University, Keelung 20224, Taiwan; chenemily0503@gmail.com (Y.-A.C.); chengw@mail.ntou.edu.tw (G.-W.C.); 2Institute of Food Safety and Risk Management, National Taiwan Ocean University, Keelung 20224, Taiwan; kuhh@email.ntou.edu.tw; 3Graduate Institute of Cancer Biology and Drug Discovery, College of Medical Science and Technology, Taipei Medical University, Taipei 11031, Taiwan; tsuichin@tmu.edu.tw; 4Graduate Institute of Medical Sciences, Department of Research and Development, National Defense Medical Center, Taipei 11490, Taiwan; hsinyi.chang@mail.ndmctsgh.edu.tw; 5Department of Nutrition &Food Science, University of Maryland, College Park, MD 20742, USA; wei@umd.edu; 6Department of Seafood Science, National Kaohsiung University of Science and Technology, Kaohsiung 811, Taiwan; yht@nkust.edu.tw

**Keywords:** *Listeria monocytogenes*, high-pressure processing, quantitative proteomics, translation initiation, stress responses

## Abstract

**Simple Summary:**

High-pressure processing (HPP) is a prevailing non-thermal food preservation technology. The inactivation mechanisms of *Listeria monocytogenes* under sub-lethal to lethal damage by different levels of HPP treatments were conducted by label-free quantitative proteomic analysis. HPP might promote translation initiation due to upregulation of most ribosomal subunits and initiation factors. However, protein synthesis was arrested according to the shortage of proteins responsible for elongation, termination and recycling. The quantitative proteomics approaches provide fundamental information on *L. monocytogenes* under different HPP pressures, and provide theoretical support for HPP against Listeriosis illness and for promotion of safer ready-to-eat foods.

**Abstract:**

High-pressure processing (HPP) is a prevailing non-thermal food preservation technology. The inactivation mechanisms of *Listeria monocytogenes* under HPP at 200 and 400 MPa for 3 min were investigated by label-free quantitative proteomic analysis and functional enrichment analysis in the Kyoto Encyclopedia of Genes and Genomes. HPP treatment at 400 MPa exhibited significant effects on proteins involved in translation, carbon, carbohydrate, lipid and energy metabolism, and peptidoglycan biosynthesis. HPP increased most ribosomal subunits and initiation factors, suggesting it might shift ribosomal biogenesis to translation initiation. However, protein synthesis was impaired by the shortage of proteins responsible for elongation, termination and recycling. HPP stimulated several ATP-dependent Clp proteases, and the global transcriptional regulator Spx, associating with activation of the stress-activated sigma factor Sigma B (σ^B^) and the transcriptional activator positive regulatory factor A (PrfA) regulons. The quantitative proteomics approaches provide fundamental information on *L. monocytogenes* under different HPP pressures, and provide theoretical support for HPP against Listeriosis illness and for promotion of safer ready-to-eat foods.

## 1. Introduction

There are approximately 48 million cases of foodborne diseases annually and foodborne listeriosis is among the most serious and severe infections caused by the bacteria *Listeria monocytogenes*. The U.S. Centers for Disease Control and Prevention estimate that approximately 1600 people become sick and 94% will be hospitalized yearly [1]. *L*. *monocytogenes* can contaminate various food, including raw meat, seafood and vegetables, ready-to-eat (RTE) processed foods, and unpasteurized milk or milk products. The U.S. Department of Agriculture’s Food Safety and Inspection Service (FSIS) established a zero-tolerance policy for *L. monocytogenes* in RTE meat and poultry products [2]. In the European Union, 5175 cases of listeriosis, more than 2 fold the cases in 2018, and the highest case fatality rate (17.6%) among foodborne diseases were reported according to the 2019 zoonoses report. A significant increase in confirmed listeriosis cases in the European Union was also observed from 2009 to 2018 [3].

High-pressure processing (HPP) treatment is among the most established non-thermal pasteurization techniques for food preservation by inactivating vegetative putrefaction and pathogenic bacteria with minimal degradation of the nutritive and organoleptic quality of food [4,5]. Application of HPP to food is mainly at a range of 100–800 MPa. The primary target of HPP is the denaturation of proteins and enzymes, the disruption of membrane and ribosomes, and the leakage of intracellular material when the pressure is higher than 400 MPa. HPP can eliminate *L. monocytogenes* in several types of meat products and partially reduce nitrites without causing negative effects on the safety of RTE dry-fermented sausages [6]. HPP can extend the shelf life of rehydrated saltfish from 5 days to more than 49 days [7]). HPP can also inactivate microbial loads and retard total volatile basic nitrogen (TVB-N) production to better preserve Asian hard clams during refrigerated storage with pressures of 400–600 MPa for 3 min [8].

The mass spectrometry (MS)-based proteomics approach comprehensively enables the identification, quantification, and characterization of global proteomes and post-translational modifications [9]. The most substantial developments in sensitivity and quantification can be attributed to significant advances in nano-UPLC, fast-scanning high-resolution accurate MS, and bioinformatics analysis methods [10]. Gene set enrichment analysis of a barotolerant *L. monocytogenes* strain S2542 indicated that HPP (400 MPa/5 min or 600 MPa/5 min) induced a significant reduction in the expression of the PrfA and SigB regulons [11]. PrfA is the primary virulence regulator, and SigB (σ^B^), in concert with PrfA helps regulate virulence in *L. monocytogenes.* Hence, adequate HPP exposure reduces virulence-related functional genes, including *hly*, *actA,* and invading surface proteins internalins A, B, G and H. However, HPP demonstrated increased expression of numerous DNA repair genes, transcription-associated genes, including RNA polymerase α subunit, delta factor, and those encoding proteins involved in transcription termination/antitermination activities (e.g., *nusA*, *nusG* and *rho*) as well as translation-apparatus-related genes, consisting of genes involved in the assembly, maturation and stabilization of ribosomal protein, tRNA and rRNA. Moreover, HPP assisted *L. monocytogenes* in translation, with 46–58 ribosome and translation-associated genes significantly upregulated and merely 3–7 genes downregulated. The expression of two cold-shock protein genes, *cspB* and *cspL*, was greatly increased, but certain heat-shock protein genes (*groEL*, *htrA*, *dnaK*, *clpC* and *clpE*) were repressed and the expression of others (*clpQ*, *clpP* and *clpX*) was only slightly increased under HPP stress. HPP also stimulated an increase in protein channel complexes, motility and chemotaxis, and lipid and cell wall biogenesis [11]. More recently, the gene expression profiling of *L. monocytogenes* strains (barotolerant RO15 and barosensitive ScottA) by HPP (200 and 400 MPa/ 8 min) damage recovery at cooling temperature (8 °C) was investigated [12]. HPP induced protein folding, the phosphotransferase system, and cobalamin biosynthesis, resulting from SigB-initiated general stress response. Cell division-associated genes were downregulated, but peptidoglycan synthesis genes were upregulated, inferring that cell wall repair played a role in HPP damage recovery. Furthermore, the ncRNAs Rli47 and Rli53 might also play a role in HPP damage recovery [12].

Label-free quantitative proteomics to explore *L. monocytogenes* inactivation mechanisms under HPP remains lacking. Limited quantitative proteomics studies have successfully applied to discover the effects of combining HPP and electrolyzed water to inactivate *Bacillus cereus* spores [13], *L. monocytogenes* from our previous work [14], *Campylobacter jejuni* 81–176 [15], and *Lactobacillus sanfranciscensis* [16]. The objective of the present study was to evaluate the efficacy and mechanism of HPP at 200 and 400 MPa in inactivating *L. monocytogenes* (from sub-lethal to lethal injuries) by label-free quantitative proteomics.

## 2. Materials and Methods

### 2.1. Bacterial Strain and Culture Conditions

The bacterial strain *L. monocytogenes* BCRC 15408 (serotype 4b) was obtained from the Bioresource Collection and Research Center (Food Industry Research and Development Institute, Hsinchu, Taiwan). The strain was routinely cultivated on tryptic soy agar (Difco Lab, Becton-Dickinson Diagnostic Systems, Sparks, NJ, USA) at 37 °C for 24 h. For inoculation, the respective pre-cultures were then transferred to a tryptic soy broth (TSB) in two successive 24 h intervals before further use.

### 2.2. High-Pressure Processing Conditions

*L. monocytogenes* was cultured in TSB at 37 °C for 18–24 h. The logarithmic growth period of bacterial culture was added to distilled water (1:19), packed and sealed in a vacuum bag (Nylon/PE), precooled to below 20 °C, and laid into an HPP600MPA 6.2-L high-pressure unit (Bao Tou KeFa High Pressure Technology Co., Ltd., Bao Tou, Inner Mongolia, China). The compression rate was 3.4 MPa/s. The temperatures increased at 6–12 °C for 200–400 MPa. The pressure levels were maintained at 200–400 MPa for 3 min and released immediately (<2 s). The same bacterial culture without HPP was used as a control. Cell viability was determined by performing serial 10-fold dilutions and using a pour plate technique on tryptic soy agar for 24 h at 37 °C.

### 2.3. Whole-Protein Extraction for Proteomic Analysis

The bacterial culture (8 log CFU/mL) with a 19-fold volume of sterile distilled water was treated by HPP at 200–400 MPa. After PBS washing and centrifugation, the treated bacterial pellets were suspended in a lysis buffer (12 mM sodium deoxycholate, 12 mM sodium N-lauroylsarcosinate, and 10 mM Tris; pH 9.0), the cells were mixed with 1 mm ceramic beads and 0.1 mm glass beads, lysed with 5 cycles at 1 min and at 6.5 m/s speed using a cell homogenizer (FastPrep-24; MP Biomedicals, LLC, Irvine, CA, USA), allowing to stand for 5 min on ice between each cycle [8]. After centrifugation (15,000× *g*, 20 min at 4 °C), the supernatant was collected as the whole-protein extract and stored at −80 °C for further use. The protein concentration was determined by the BCA Protein Assay Kit (Thermo Scientific, Rockford, IL, USA) according to manufacturer instructions. The control group was labelled as C, and samples treated with HPP of 200, 300 and 400 MPa were labelled as H2, H3 and H4, respectively.

### 2.4. Mass Spectrometric Analysis

#### 2.4.1. In-Solution Digestion

The whole-protein extracts were reduced using 10 mM dithiothreitol (DTT) at 37 °C for 30 min, and then alkylated with 50 mM iodoacetamide (IAA) at 37 °C for 30 min in the dark. Enzymes were digested with Lys-C/50 mM ammonium bicarbonate (ABC) followed by trypsin/50 mM ABC at 37 °C for 16 h. Trifluoroacetic acid (TFA) was added to a final concentration of 1% to stop the enzyme digestion and an equal amount of ethyl acetate was added to extract detergent. The mixture was shaken for 2 min, and centrifuged at 17,000× *g* for 2 min. Then, the organic phase of the mixture was removed, and the aqueous layer was dried using Speed Vac at 50 °C. The dried samples were redissolved in 0.1% TFA, and desalted using Zip-Tip according to manufacturer instructions.

#### 2.4.2. Liquid Chromatography Coupled Tandem Mass Spectrometry (LC–MS/MS)

Peptide mixtures were analyzed using an LTQ Orbitrap Velos^TM^ Hybrid Ion Trap–Orbitrap Mass Spectrometer (Thermo Scientific, Rockford, IL, USA). The peptide mixtures were loaded onto a nanoACQUITY Ultra Performance LC (Waters, Milford, MA, USA) with a C18 BEH column (ID 75 μm; length 25 cm; Waters) packed with 1.7 μm particles with a pore size of 130 Å. Analytes were separated using a segmented gradient within 90 min from 5% to 40% solvent B (acetonitrile with 0.1% formic acid) at a flow rate of 300 nL/min and a column temperature of 35 °C. Solvent A was 0.1% formic acid in water. The effluent from the UPLC column was directly electrosprayed onto the mass spectrometer. The mass spectrometer was operated in the positive ion mode for 120 min, and a data-directed analysis method (*m*/*z* 350–1600, 1 s/scan) was employed. In each cycle, a full-scan spectrum was acquired with resolution R = 100,000, followed by collision-induced dissociation on the three most intense ions with a target range of *m*/*z* 100–2000 (1 s/scan).

### 2.5. Data Analysis and Protein Network Construction

Three independent biological replicates were performed for each experiment. LC–MS/MS data were analyzed using the Maxquant software (v. 1.6.6.0; Max Planck Institute of Biochemistry, Martinsried, Germany) for label-free quantitation (LFQ). Peptides and proteins were identified using a false discovery rate of 1%. Protein sequence information on *L. monocytogenes* serovar 1/2a was obtained from UniProtKB (downloaded on 11 December 2018). LFQ proteomic analysis of three biological replicates was performed to explore the mechanism of global proteome changes in *L. monocytogenes* under water-washed (C), HPP-200 MPa (H2) and HPP-400 MPa (H4) treatments. Differentially expressed proteins (DEPs) were defined if a 2-fold change was observed in expression levels between control and HPP treatment groups that was statistically significant (*p* < 0.05) [13] using Perseus software (v. 1.6.7.0; Max Planck Institute of Biochemistry, Martinsried, Germany). The functional annotation tool The Database for Annotation, Visualization, and Integrated Discovery (DAVID, v. 6.8) was employed using gene ontology (GO) and UniProtKB Keywords, STRING Protein–Protein Interaction Networks (v. 11.0) [17], the eggnog online framework of Database of Clusters of Orthologous Groups of proteins (COGs), and the Kyoto Encyclopedia of Genes and Genomes (KEGG) to determine the protein functions [18]. The dose-dependent effects of HPP in *L. monocytogenes* proteomes were identified from three independent samples, as shown in Appendix A.

### 2.6. Machine Learning of Grouping Proteins

#### 2.6.1. The Data Manipulation Phase

Figure 1 illustrates the detailed flowchart of the *k*-means clustering algorithm, including data manipulation, analysis and clustering phases. The data manipulation phase is responsible for collecting and performing data normalization of row data. This study sets seventeen fields, which are Protein ID, Name, Entry Name, Gene, Locus, C-1, C-2, C-3 and log_2_ (fold change): (H2-1/C-1), (H2-2/C-2), (H2-3/C-3), (H3-1/C-1), (H3-2/C-2), (H3-3/C-3), (H4-1/C-1), (H4-2/C-2) and (H4-3/C-3). The C stands for control. The H2, H3 and H4 are HPP at 200, 300 and 400 MPa, respectively. The three independent biological replicates were conducted for each group.

#### 2.6.2. The Data Analysis Phase

The data analysis phase is responsible for determining the optimal *k* clusters. This study used the Gap Statistic to estimate the best number of clusters. Every gap is a logarithmic difference between the mean dispersion of reference datasets and the dispersion of the original dataset [19]. The detailed computations of the Gap Statistic are described as follows.
(i)To cluster observed proteins. Varying the total number of clusters from *k* = 1, 2, …, *K*, and giving within-dispersion measures *W_k_*, where *k* = 1, 2, …, *K*. *E*(*W_k_*) is the expected value of total variation within-group variance.(ii)To generate *B* reference datasets and to cluster each one, giving the within-dispersion measure *W_kb_*, where *b* = 1, 2, …, *B* and *k* = 1, 2, …, *K*. The Gap Statistic, *Gap*(*k*), is estimated using Formula (1)
(1)$$Gap\left(k\right)=E\left(W_{k}\right)-W_{k}=\frac{1}{B}\overset{B}{\underset{b=1}{\sum }}\log\left(W_{kb}\right)-\log\left(W_{k}\right)$$(iii)To compute the standard deviation (*Sd*(*W_k_*)) and the standard error (*s_k_*). The standard deviation (*Sd*(*W_k_*)) and standard error (*s_k_*) are described using Formulae (2) and (3).
(2)$$Sd\left(W_{k}\right)=\sqrt{\frac{1}{B}\overset{B}{\underset{b=1}{\sum }}{\left(\log\left(W_{ki}\right)-E\left(W_{k}\right)\right)}^{2}}$$
(3)$$s_{k}=Sd\left(k\right)*\sqrt{1+\frac{1}{B}}$$(iv)To choose the optimal number of clusters. Hence, *k** is estimated as the smallest *k*, such that *Gap*(*k*) ≥ *Gap*(*k* + 1) *− s_k+1_.*


#### 2.6.3. The Data Clustering Phase Is Responsible for Setting Clusters and Separate Proteins

The k-means clustering algorithm is to cluster the proteins into *K* pre-defined distinct non-overlapping clusters, *k**. Three steps are described as follows.
(i)Initialization of centroid.

Given a protein dataset *P* = {*p*_i_|*i* = *1,…n*}, k-means select *k** initial seeds.

(ii)Group proteins to centroid *k** based on minimum distance.

The algorithm divides the set of proteins into *k** clusters by the minimum sum of distance with the least squared Euclidean distance. k-means clustering aims to partition the *n* proteins into *k** sets. Hence, to minimize the within-cluster sum of squares (*S_i_*), we use Formula (4)
(4)$$S_{i}=argmin\overset{k^{*}}{\underset{i=1}{\sum }}\overset{}{\underset{P\in S_{i}}{\sum }}{\left|\left|P-\mu_{i}\right|\right|}^{2}\;$$
where *μ_i_* is the mean of points in *S_i_*. 

(iii)Update centroids.

Recalculate means of *μ_i_* and then assign to each cluster, which is described as Formula (5).
(5)$$\mu_{i}^{t+1}=\frac{1}{S_{i}^{t}}\underset{p_{j}\in S_{i}^{t}}{\sum }p_{j}$$
where *t* is the *t*th operation. When $S_{i}^{t+1}=S_{i}^{t},\;\forall i=1,\ldots ,k$, the k-means algorithm has converged.

There are 4 different evaluations, which are the 9G, 9GE, 3G and 3GE groups. Related descriptions are described as follows.
(1)9G: log_2_ (H2-1/C-1), log_2_ (H2-2/C-2), log_2_ (H2-3/C-3), log_2_ (H3-1/C-1), log_2_ (H3-2/C-2), log_2_ (H3-3/C-3), log_2_ (H4-1/C-1), log_2_ (H4-2/C-2) and log_2_ (H4-3/C-3).(2)9GE: [log_2_ (H2-1/C-1)]/2, [log_2_ (H2-2/C-2)]/2, [log_2_ (H2-3/C-3)]/2, [log_2_ (H3-1/C-1)]/3, [log_2_ (H3-2/C-2)]/3, [log_2_ (H3-3/C-3)]/3, [log_2_ (H4-1/C-1)]/4, [log_2_ (H4-2/C-2)]/4 and [log_2_ (H4-3/C-3)]/4. Herein E stand for equivalence, and 9GE mimic log_2_ (fold change) under 100 MPa equally for each group.(3)3G: mean values of log_2_ (H2/C), log_2_ (H3/C), and log_2_ (H4/C).(4)3GE: mean values of [log_2_ (H2/C)]/2, [log_2_ (H3/C)]/3, and [log_2_ (H4/C)]/4.

## 3. Results

### 3.1. The Influence of HPP on L. monocytogenes Viability

HPP pressures of 100 to 400 MPa for HPP (treatment duration 3 min) were selected to determine the bactericidal activity of the different levels of HPP treatments on *L. monocytogenes* (Appendix A). The initial bacterial number was approximately 7 log CFU/mL. HPP at 100, 200 and 300 MPa resulted in 7.35 ± 0.07, 7.03 ± 0.18 and 5.51 ± 0.08 log CFU/mL survival, respectively. HPP at 400 MPa completely inactivated bacterial growth. Hence, we selected HPP pressures of 200 and 400 MPa (sub-lethal to lethal damage) to examine the inactivation mechanism using the LFQ proteomics approach.

### 3.2. The Influence of HPP on Differential Expression of L. monocytogenes Proteomes

The Pearson correlation coefficients for the same treatment among the three replicates were 0.965–0.986. The more differentially expressed patterns were obtained when the control was compared with the H4 or H2, for which the coefficients were 0.879–0.900 and 0.904–0.921, respectively. The H2 and H4 correlation coefficients were as high as 0.952–0.972, implying that 200 and 400 MPa only caused marginal changes in *L. monocytogenes* proteomes (Appendix A).

In total, 380 proteins were quantitated with high confidence and proteins with fold changes of ≥ 2 and *p* < 0.05 in comparison with the control group were considered to be the differentially expressed proteins (DEPs). The total number of DEPs was 56 and 73 for H2/C and H4/C, respectively (Appendix A). The number of upregulated DEPs was 39 and 46 for H2/C and H4/C, respectively (Appendix A).

The number of downregulated DEPs for H2/C and H4/C was 17 and 27, respectively (Appendix A). According to volcano plot analysis, upregulated DEPs exhibited a greater number than downregulated DEPs in H2/C and H4/C (Figure 2a). The DEPs accounted for 14.7 and 19.2% of total identified proteins for H2/C and H4/C, respectively. Six DEPs, namely 30S ribosomal protein S21 (H2/C: 3.41; H4/C: 4.52 log_2_ fold), translation initiation factor IF-3 (H2/C: 3.75; H4/C: 4.08), 50S ribosomal protein L24 (H2/C: 3.36; H4/C: 4.08), 30S ribosomal protein S13 (H4/C: 3.90), 30S ribosomal protein S12 (H4/C: 3.47) and 30S ribosomal protein S20 (H4/C: 3.42), had more than 10-fold changes (equal to log_2_ value 3.32). Four DEPs, namely D-alanyl carrier protein (H4/C: −5.20), UPF0473 protein lmo1501 (H2/C: −3.69; H4/C: −3.50), 50S ribosomal protein L33-1 (H2/C: −4.27) and ribosome-binding factor A (H2/C: −4.10), had less than 0.1-fold changes (equal to log_2_ value −3.32) (Figure 2a and Table 1).

### 3.3. Machine Learning of Grouping DEPs

This section is on producing groups of 79 proteins with a high degree of similarity within each group and a low degree of similarity between groups by using an unsupervised *k*-means clustering learning algorithm (Figure 1). Four clusters are the optimal number of clusters after Gap Statistic analysis, where the smallest *k* was determined by *Gap*(*k*) ≥ *Gap*(*k* + 1) − *s*_k+1_ (Appendix A). After determining the optimal number of optimized clusters *k* = 4, the *k*-means was applied to divide 79 proteins into 4 clusters (Appendix A). The detailed characteristics of grouping DEPs are listed in Appendix A, where four different evaluations are group 9G, 9GE, 3G and 3GE.

The 48 upregulated DEPs were classified into four groups according to the hierarchical clustering heat maps: UI: 8 DEPs including rpsT, rpsL, rpsM, rpmF2, rpmA, rplX, rpsU and infC; UII: dapF; UIII: 10 DEPs including luxS, rpsJ, rplO, rplF, rpmD, rplI, rpsI, rpsP, rpsG and rpsS; UIV: the other 29 DEPs (Figure 2b). The 8 DEPs of the group UI exhibited apparent up-regulated (>3.0 log_2_ (fold change)). rpsU, infC and rplX was upregulated by 4.0 folds under H4/C treatments. dapF (group UII) was upregulated in decreasing order (1.78 for H2/C and 1.28 for H3/C) but for H4/C was slightly downregulated (−0.41). Except for rpsI, rpsJ and luxS, 7 out of 10 DEPs in the group UIII revealed an increasing order of upregulation—mostly within the range from 2 to 3 log_2_ (fold change). All the 29 DEPs of the group UIV showed lower levels (<2.0) of upregulation except for tal1 (2.64) and rplR (2.10) under H4/C treatment

On the other hand, the 48 upregulated DEPs were classified into 2 clusters according to k-means clustering: 16 DEPs for cluster 1 (HUR) and 32 DEPs (UR) for cluster 2 (Figure 2b and Appendix A). All treatments of 16 DEPs exceeded 2.0 log_2_ (fold change) except for rpmD (1.97) and rplO (1.89) under H2/C treatments. The DEPs of HUR cluster were the same as the hierarchical clustering group UI and UIII excluding luxS and rpsJ. UR (cluster 2) had 32 DEPs in the hierarchical clustering group UII, UIV, luxS and rpsJ. There are four different evaluations, 9G, 9GE, 3G and 3GE groups, depending on each repetition and its mean values. All four clusters (HUR, UR, DR and HDR) had the same numbers of DEPs under the four evaluations except for rplO. rplO represented clusters 1 and 2 under 9G/9GE/3G, and 3GE evaluations, respectively. The DEPs of cluster 1 (HUR) in group 3G had a greater than 2.0 log_2_ fold change under both H3/C and H4/C treatments, whereas the DEPs, including rplO, of cluster 2 at 3GE had a less than 1.0 log_2_ fold change.

The 31 downregulated DEPs were classified into four groups according to the hierarchical clustering heat maps: DI: rpmG1; DII: dltC and lmo1501; DIII: rbfA; DIV: the other 27 DEPs (Figure 2b). rpmG1 (in group DI) was notably different from other downregulated DEPs that exhibited apparent downregulated in H2/C and H3/C more than −4 but showed slightly upregulated in H4/C. The dltC showed in a dose-dependent fashion that H3/C and H4/C had a −3.71 and −5.20 log_2_ fold change, respectively. lmo1501 showed a −3.69, −2.92 and −3.50 log_2_ fold change in H2/C, H3/C and H4/C, respectively. dltC and lmo1501 were classified into the group DII. rbfA (group RIII) had only one high level of downregulation (E2/C, −4.10 log_2_ fold change). The DEPs of the group DIV had lower log_2_ fold changes (<−3.50) under all HPP treatments. 

On the other hand, the 31 downregulated DEPs were classified into two clusters according to k-means clustering: 28 DEPs in cluster 3 (DR) and three DEPs, rpmG1, rbfA and lmo1501, in cluster 4 (HDR) (Figure 2b and Appendix A). The DEPs of cluster 3 mostly revealed a dose-dependent pattern and a lower than −3.50 log_2_ fold change except for dltC (−0.84 for H2/C, −3.71 for H3/C and −5.20 for H4/C). All three DEPs of cluster 4 had a higher log_2_ fold change for H2/C (>−3.50). Hence, the DEPs of cluster 4 were also relevant to RI, RII (lmo1501) and RIII according to the hierarchical clustering heat maps.

### 3.4. GO and COGs Enrichment Analysis of HPP-Induced DEPs

Functional classification was used to characterize terms enriched in gene ontology biological processes (GOBPs), molecular functions (GOMFs), and cellular components through GO analysis and string functional protein association networks. Considering treatment with low-pressure HPP (E2/C), there were 56 DEPs comprising 50 proteins annotated in 8 GOBPs, primarily cellular metabolic (49) and biosynthetic processes (45), followed by protein metabolic process (36 blue nodes), translation (34 red nodes), the nucleic acid metabolic process (12 green nodes), ribosome biogenesis (8 yellow nodes), biological regulation (6 dark green nodes), and DNA-templated transcription (4 pink nodes) (Figure 3a). Regarding GOMFs, 30 proteins showed structure molecule activity for ribosomes and 31 participated in nucleic acid binding, including 4 in DNA binding, and 19, 5, and 3 proteins in rRNA, tRNA, and mRNA binding, respectively. There were 45 proteins in the cytoplasm, with 30 ribosomal proteins. Under high pressure, H4/C treatment exhibited all 73 DEPs for 11 GOBPs, primarily cellular processes (65) and cellular metabolic processes (63), including 47 and 16 proteins for macromolecule and small-molecule metabolic processes, respectively. Among them, 37 DEPs were involved in the protein metabolic process (blue nodes) and 35 DEPs were involved in translation (red nodes). Other DEPs related to the nucleic acid metabolic process (15 green nodes), ribosome biogenesis (8 yellow nodes), biological regulation (8 dark green nodes), and DNA-templated transcription (6 pink nodes) were also indicated. The other 3 GOBPs specific for the H4/C group were the carbohydrate derivative biosynthetic process (8 light blue nodes), peptidoglycan-based cell wall biogenesis (3 orange nodes), and nucleotide-excision repair (2 purple nodes (Figure 3b). The abundant GOMFs involved 46 DEPs responsible for binding and 34 proteins for nucleic acid binding, including 6 for DNA binding and 28 for RNA binding (20, 5, and 3 proteins for rRNA, tRNA and mRNA binding, respectively). The 27 proteins exhibited catalytic activity. In total, 58 proteins were related to cellular components, mainly involved in the ribosome. Overall, H4/C exhibited similar but enhanced effects with H2/C on GOBPs, GOMFs and cellular components.

According to the Database of Clusters of Orthologous Groups of proteins (COGs), most DEPs with strong interactions (64.3 and 52.1% for H2/C and H4/C, respectively) were related to translation, ribosomal structure and ribosome biogenesis (32 ribosomal proteins, ribosome-recycling factor, phenylalanine-tRNA ligase alpha subunit, translation initiation factor IF-3, D-alanyl carrier protein, elongation factor P, aspartyl/glutamyl-tRNA (Asn/Gln) amidotransferase subunit C, ribosome-binding factor A, ribosome maturation factor RimM, ribonuclease Z and threonine-tRNA ligase) (Appendix A). The remaining DEPs were mainly involved in nucleotide transport and metabolism (7.1 and 8.2% for H2/C and H4/C, respectively) and in transcription (7.1 and 8.2% for H2/C and H4/C, respectively) (Appendix A).

### 3.5. KEGG Pathway Analysis of HPP-Induced DEPs

KEGG collecting comprehensive and high-level functions of the biological system from the molecular-level information allows the discovery of collective biological pathways from any given set of differentially expressed genes, proteins, or metabolites. To understand the molecular regulations caused by sub-lethal and lethal effects of HPP, the DEPs derived from 200 and 400 MPa and their first-order interacting neighbor proteins were subjected to KEGG pathway analysis (Table 1 and Figure 3). The lethal and sub-lethal DEPs identified from H4 or H2 compared with the control group were mainly enriched in KEGG pathways including translation, secondary metabolite biosynthesis, and signaling (Figure 4). DEPs involved in ribosomes, secondary metabolite biosynthesis, microbial metabolism in diverse environments, amino acid biosynthesis, cofactor biosynthesis, the pentose phosphate pathway, and ABC transporters were dose dependent, while the pathways carbon metabolism, glycine, serine and threonine metabolism, D-alanine metabolism, and fatty acid metabolism were exclusively enriched on lethal dose (400 MPa) of HPP. DEPs involved in amino acid metabolism and the environmental sensing system were equally activated in both lethal (400 MPa) and sub-lethal (200 MPa) treatments, suggesting those functions might serve as the initial response for HPP (Table 1 and Figure 4).

## 4. Discussion

HPP at 200 MPa was ineffective to inactivate *L. monocytogenes* RO15 and ScottA strains, yet 400 MPa significantly reduced 5.78–7.04 log CFU/mL for RO15 and ScottA, respectively [12]. In our previous report, the combination of HPP at 300 MPa and slightly acidic electrolyzed water at 20 ppm (available chlorine concentration) could achieve complete inactivation [14]. When the pressure is between 100 and 300 MPa, the conformation changes of the proteins were reversible. Protein denaturation might be irreversible when the pressure is >300 MPa [20,21].

### 4.1. Translational Regulation

The ribosome is responsible for protein synthesis, which consists of four essential parts—initiation, elongation, termination, and recycling. Thirty-two ribosomal subunit proteins were found to have significantly upregulated expression, with only one exception, 50S ribosomal protein L33-1, which was downregulated by 4.27 fold in H2/C. However, higher hydrostatic pressure changed the significant downregulation (−4.27 for 200 MPa) of L33-1 to a minor upregulation (0.17 for 400 MPa) (Appendix A). Typically, the 32 ribosomal proteins showed a dose-dependent increase in fold change when *L. monocytogenes* was exposed to pressures of 200 and 400 MPa (Appendix A). HPP on ribosome dissociation was shown on vegetative cells of *L. monocytogenes*, *Bacillus subtilis* and *Escherichia coli* [12,22,23]. During the growth arrest phase, HPP induced a significant number (46–58) of upregulated ribosomal protein genes and translation-associated genes, namely translation initiation factors, but merely 3–7 genes were downregulated by more than 2 fold [11]. Sucrose density gradient sedimentation analysis revealed that ribosomes were dissociated in a pressure-dependent manner and then reconstructed. Transcriptome analysis using vegetative cells of *B. subtilis* also revealed that the translational machinery can preferentially be reconstructed upon the HHP 250 MPa treatment [22]. Furthermore, HPP induced dissociated free 30S and 50S ribosomal subunits, which are more susceptible to degradation by endonucleases into nucleotides than 70S ribosomes, and their ribosome-derived nucleotides could be recycled as a nutrient source for the growth of starved *E. coli* under nutrient-limiting conditions [24].

Initiation occurs only at the free 30S subunits by interacting with initiation factors (IFs) when a 30S subunit comprises mRNA transcripts, IF-3:IF-1 complex, IF-2, GTP, and a particular initiator, aminoacyl tRNA. The 30S subunits require IF-3 for mRNA binding. Meanwhile, IF-3 has to be released from 30S subunits to create the complex of 50S subunits connected with the mRNA:30S subunit. As a result, initiation is the rate-limiting step in protein synthesis, and therefore several critical translational regulations occur during initiation. This study showed that all IFs increased under the HPP 200 and 400 MPa treatments, although both the IF-1 and IF-2 levels slightly increased. Moreover, IF-3 increased by 4.08 fold under H4/C treatment (Table 1). The increases in IFs and ribosomal subunit proteins indicated that the HPP at 200 and 400 MPa treatments might promote the initiation of protein synthesis.

The essential composite for peptide chain elongation is an mRNA:70S ribosome:peptidyl-tRNA complex, aminoacyl tRNA, elongation factors, and GTP. A significant increase in IF-3 may occupy 30S subunits to prevent re-associating with 50S subunits and hinder the subsequent elongation process. tRNA is a vital adaptor molecule in protein synthesis. Specific aminoacyl-tRNA synthetases (aaRSs) produce aminoacyl tRNA by conjugating amino acids to the 3′-CAA acceptor end of tRNA. Each aminoacyl-tRNA synthetase matches up its amino acid, with tRNA having proper anticodons that can base-pair with codons from mRNA specifying the particular amino acid. The correct loading of amino acids onto various tRNA is crucial to achieving translation fidelity. The aaRSs serve a dual role in charging tRNAs, providing protein synthesis flux and reducing uncharged tRNA levels. ThrRS, responsible for aminoacylation of threonyl-tRNA, was downregulated by −1.08 log_2_ fold in H4/C, but not significant (−0.64) in H2/C (Appendix A). Moreover, AsnRS, ArgRS, IleRS, TyrRS, AspRS, ProRS, and CysRS were all downregulated but not significant in both the H2/C and H4/C conditions. The three proteins, GatC/A/B, develop correctly charged Asn-tRNA or Gln-tRNA through the transamidation of misacylated Asp-tRNA or Glu-tRNA in prokaryotes which are commonly lacking either or both asparaginyl-tRNA and glutaminyl-tRNA synthetases. GatC/A/B were downregulated; in particular, GatC significantly decreased by 2.39 and 2.18 fold under 200 and 400 MPa, respectively (Appendix A). It implied that biosynthesis of Gln and Asn was inhibited under HPP treatments. On the other hand, PheRS, GlyRS, HisRS, ValRS, TrpRS, SerRS and LeuRS were all upregulated, although only PheRS was significantly upregulated by 1.05 log_2_ fold in H2/C (Appendix A). Furthermore, this study showed that elongation factors (EFs) including EF-Tu, EF-4, EF-P and EF-G were all reduced after the HPP 200 and 400 MPa treatments, although only the results for EF-P in all groups were significant (Table 1 and Appendix A). These lines of evidence indicated that the HPP 200 and 400 MPa treatments may diminish peptide chain elongation of protein synthesis.

During termination, release factors (RFs) promote polypeptide release from 70S ribosomes. This study showed that both RF-1 and RF-3 decreased but RF-2 increased after the HPP 200 and 400 MPa treatments but without statistical significance (data not shown).

Finally, 70S ribosomes are dissociated into free 30S and 50S subunits through the cooperative action of three translation factors consisting of ribosome recycling factor (RRF), EF-G and IF-3 during the recycling of free ribosomal subunits. IF-3 is an anti-association factor, whereas RRF is a dissociation factor. Collaboratively, RRF, EF-G and GTP transiently dissociate 70S ribosomes into subunits, and IF-3 releases the deacylated tRNA from the 30S subunit, thus preventing re-association back into 70S ribosomes while running out of GTP [25]. The stable dissociation complex for 70S ribosomes into 30S/50S subunits is dependent on the formation of RRF-EF-G-IF-3. When the ribosome cycle finishes, the 30S/50S subunits are recycled for subsequent translation. This study showed that both RRF and IF-3 were elevated but EF-G decreased under HPP treatments, although only the results for IF-3 in all groups and RRF in H4/C were significant (Table 1 and Appendix A). HPP treatments might hinder the recycling of free ribosomal subunits due to EF-G shortage. Therefore, HPP seems to promote initiation but hinders the subsequent elongation, termination, and recycling steps during protein translation.

### 4.2. HPP Promoted Translation Initiation and Retarded Ribosome Biogenesis

The maturation factors, including ribosome maturation factor M (RimM), 30S ribosome-binding factor A (RbfA), ribosomal RNA small subunit dimethyl transferase A (KsgA/RsmA), and GTPase (Era), are responsible for ribosomal biogenesis. Era can bind to 16S rRNA to prevent 30S and 50S joining, facilitating rRNA maturation and 30S ribosome biogenesis. RsmA, ribosome biogenesis factor, binds to a pre-30S particle and specifically dimethylates two conserved adenosines in the terminal of 16S rRNA helix [26]. RbfA is associated with 16S rRNA but not 70S ribosomes or polysomes, and is essential in late-30S ribosomal subunit maturation [27]. Recently, RbfA and RimM were found to be involved in two distinct 30S assembly stages consisting of the early development of the central pseudoknot with the folding of the head and further docking of helix 44 to form important inter-subunit bridges in the mature structure at a later stage [28]. A comprehensive review revealed that increased RbfA synthesis could moderately attenuate △*rimM* mutation. The overexpression of Era and RsmA proteins can repress △*rbfA* mutation and cold-sensitive *Era* mutant, respectively. This implied a functional order for the four proteins, RimM, RbfA, Era, and RsmA, with RsmA as perhaps the final protein to complete 30S ribosome assembly in bacteria [29]. This study showed that RimM (−0.70 in H2/C and −1.09 in H4/C), RbfA (−4.10 in H2/C and −0.87 in H4/C), Era (−0.34 in H2/C and −0.46 in H4/C), and RsmA (−0.64 in H2/C and −0.88 in H4/C) were all reduced under the 200 and 400 MPa treatments, although the result was only significant for RimM in H4/C, and RbfA in H2/C (Appendix A). The decreases in ribosomal maturation factors indicated that HPP at 200 and 400 MPa might hinder ribosome maturation and biogenesis.

The latest study showed that RbfA in the immature *E. coli* 30S subunit could promote ribosome biogenesis and suppress protein synthesis; instead, RbfA is displaced by IF-3 after 30S maturation initiates translation [29]. In this study, IF-3 was markedly upregulated, and RbfA was downregulated. This might indicate that abundant IF-3 could substitute RbfA to bind with 30S ribosome, facilitating translation initiation for better adaption to nutrition starvation, extreme temperature, and antibiotic stress. Hence, loss of RbfA and elevated IF-3 during HPP treatments might maintain the fidelity of bacterial protein synthesis for better tolerability to HPP treatments.

Moreover, the shortage of charged aminoacyl tRNA stringently induces the accumulation of ribosomes stalled with non-aminoacylated (uncharged) tRNA, activating the production of guanosine tetraphosphate (ppGpp) and guanosine pentaphosphate (pppGpp) to terminate rRNA transcription and suspend ribosome assembly [30]. This study revealed that aaRSs comprising AlaRS, LysRS, ThrRS, IleRS, ArgRS, TyrRS, ProRS, AsnRS, CysRS and AspRS, as well as the GatC/A/B system, were all reduced in H4/C, although the findings were significant only for ThrRS in H4/C and GatC in H2/C and H4/C (Table 1 and Appendix A), implying that HPP inhibits ribosome assembly. HPP induced ribosome dissociation, resulting in the abundant ribosomal subunit protein, and thus accelerated feedback regulation to hinder ribosome biogenesis [11,30]. This study showed that 32 ribosomal subunit proteins were significantly increased (Figure 1, Appendix A), indicating that HPP may moderate ribosome biogenesis.

Finally, the rRNA operon (*rrn*) anti-termination system is vital in rRNA synthesis, folding, and ribosome activity. Nus proteins can topologically constrain the 16S rRNA in a growing loop that supports standard rRNA folding and processing. The Nus-modified transcription complex in rrn operons serves as an RNA chaperone for assisting 16S rRNA folding and RNase III processing, resulting in the formation of functional 30S ribosome subunits [31]. The NusB proteins shown in this study were significantly downregulated (−1.15 and −1.31 log_2_ fold in H2/C and H4/C, respectively) (Table 1 and Appendix A), again indicating that HPP may weaken ribosome biogenesis. These lines of evidence confirm that HPP may reduce ribosome biogenesis and promote translation initiation.

### 4.3. HPP Response-Associated Pathways

Proteolysis and redox status both contribute to *L. monocytogenes* virulence, environmental stress adaption and the multifaceted interactions among the class three stress gene repressor (CtsR), the alternative general stress sigma factor (σ^B^), and the positive regulatory factor A (PrfA) regulons. *L. monocytogenes* expresses several ATPases, including ClpC, ClpP and ClpE, associated with diverse cellular activities. ClpP is found in many prokaryotic cells and is often involved in stress response and virulence by functioning as a molecular chaperone [32]. An inconsistency of CtsR or Clp proteins in a cell may greatly damage the cell because Clp proteins eradicate misfolding proteins and regulate many critical cellular processes such as short-lived regulators. CtsR represses class III heat shock genes, including *clpP*, *clpE*, and the *clpC* operon [33]. The high HPP-tolerant *L. monocytogenes* ScottA was established when overexpressing *clp* genes with an inactive CtsR repressor by deleting a codon in the glycine-rich domain (*ctsRΔGly*) [34]. ClpP and ClpC are both transcribed from σ^B^-dependent promoters. This study revealed that HPP induced ClpP significantly and ClpX slightly, possibly facilitating protein folding and maturation under HPP stress (Table 1, Appendix A). Except for the proteolytic core ClpP, the different cognate Clp-ATPases fine-tune the proteolytic response. ClpY (also called ATP-dependent protease ATPase subunit HslU) and ClpQ (ATP-dependent protease ATPase subunit HslV) were also slightly upregulated by 0.65 and 0.47 log_2_ fold in H4/C, respectively. However, ClpB were slightly downregulated under the HPP 200 and 400 MPa treatments. ClpB, controlled by CtsR, induces thermal tolerance and virulence but not general stress tolerance in *L. monocytogenes* [35]. Similar results for *Lactobacillus sanfranciscensis* showed that only HPP but not other stress such as extreme temperatures, salt, acid or starvation could induce ClpP protein [36]. Since stress-induced ClpP promotes the intracellular survival of *L. monocytogenes* by modulating the presentation of protective antigens listeriolysin O, ClpP may represent a potential vaccine target against *L. monocytogenes* [37]. Taken together, we suggest that HPP induced the expression of the ubiquitous Clp protease system consisting of ClpP, ClpX, ClpY and ClpQ for better adaptation upon HPP stress.

PrfA requires binding a signaling molecule, glutathione (GSH), to an entire active state. GSH is synthesized by the glutathione biosynthesis bifunctional protein (GshAB), and regulated by the GSH reductase and global transcriptional regulator (Spx), a thio-sensing and putative regulator of GSH biosynthesis [38]. The promoters of GSH reductase and Spx transcription are σ^B^ dependent. GSH could be oxidized to GSSG, which can be reduced to GSH by GSH reductase. Furthermore, the downregulation of Spx resulting in lower PrfA regulon expression is controlled directly by σ^B^ [39]. A feedback regulation loop is operated by the Spx, ClpXP and CtsR regulon [32]. Spx is also a key target of the ClpXP protease complex in *Bacillus subtilis*. Spx contributes to CtsR regulon activity and directly mediates *clpX* expression. Hence, σ^B^ could affect PrfA activity through modulation of GSH homeostasis and Spx activity. The serine-protein kinase RsbW, an anti-sigma factor, inactivates its specific antagonist protein RsbV through phosphorylation. RsbW is released upon phosphorylation of RsbV and negatively regulates σ^B^ activity by blocking its ability to form an RNA polymerase holoenzyme [40]. Spx significantly increased by 1.02 and 1.73 fold in H2/C and H4/C, respectively (Table 1 and Appendix A). GshAB and σ^A^ also increased, whereas RsbW decreased but not significantly (data not shown). These lines of evidence indicated that HPP induced σ^B^, Spx, GshAB and ClpXP, and possibly PrfA regulon activation. They also inferred that HPP may activate the σ^B^-mediated general stress response of *L. monocytogenes* although the enzyme activities of GshAB, GSH reductase, and Clp protease need to be further investigated.

Quorum sensing, also under σ^B^ control, is an interspecies bacterial communication that enables bacteria to cooperatively adapt behavior through the changes in the cell biomass and bacterial community composition. The autoinducers responsible for quorum sensing are the extracellular and small diffusible signal molecules secreted by both Gram-positive and Gram-negative bacteria. More recently, quorum sensing makes microbial density in the gastrointestinal mucosal interface and plays a key role for host health and disease [41]. These molecules are used for ‘quorum sensing’ of bacteria to synchronize biofilm formation, motility, invasion, virulence, survival, and metabolism. Autoinducer-2 (AI-2) is produced from S-adenosyl methionine (SAM) through the activated methyl cycle and by the enzyme LuxS, a broadly distributed signal molecule across bacteria. Lacking LuxS could result in reduced growth due to defective signaling, methionine recycling or accumulation of intermediates of SAM metabolism, quorum sensing, virulence, or biofilm formation [42].

On the contrary, the LuxS/AI-2 system does not function as a quorum-sensing molecule in *Campylobacter jejuni* and *Staphylococcus aureus* under specific conditions. Amino acid biosynthesis, quorum sensing, and the reduced phosphorylation level were well correlated with spore inactivation of SAEW under HPP conditions on *Bacillus cereus* spores, according to KEGG pathway analysis [13]. LuxS significantly increased (1.93~1.99 log_2_ fold) under the HPP 200 and 400 MPa treatments (Table 1 and Appendix A), inferring that HPP promotes quorum sensing and biofilm formation.

## 5. Conclusions

This study explored the quantitative changes in *L. monocytogenes* proteomes after HPP treatment for 3 min. Only HPP at 400 MPa could completely inhibit bacterial growth. According to COG clustering, the biofunctions of DEPs upon 400 MPa (lethal injury) were associated with cell cycle control, cell division, and chromosome partitioning; cell wall, membrane, and envelope biogenesis; energy production and conversion; lipid transport and metabolism; nucleotide transport and metabolism; post-translational modification, protein turnover and chaperone; replication, recombination and repair; transcription as well as translation, ribosomal structure and biogenesis (Appendix A). The information generated from this study achieves a better understanding of the bacterial deactivation under different levels of HPP, hence facilitating the development of intervention strategies in preventing *L. monocytogenes*-borne illness.

## Figures and Tables

**Figure 1 biology-11-01152-f001:**
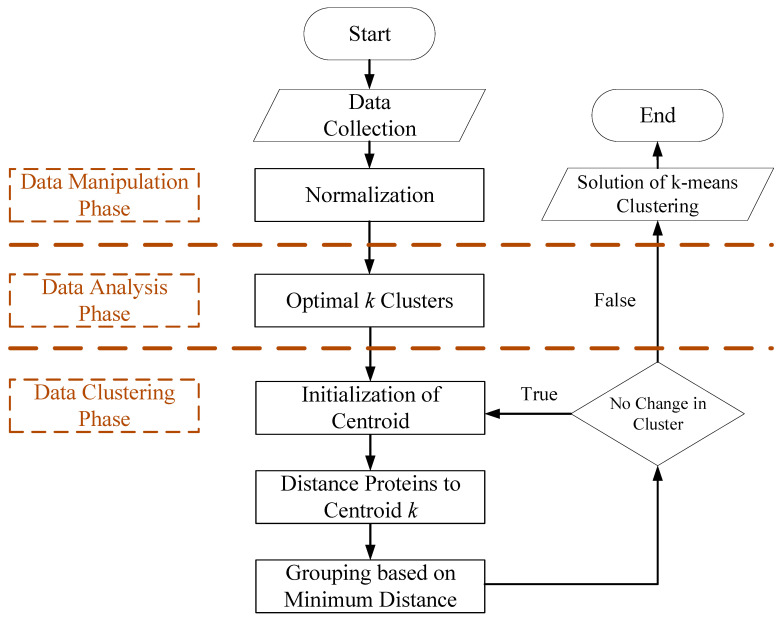
Flowchart of the *k*-means clustering algorithm.

**Figure 2 biology-11-01152-f002:**
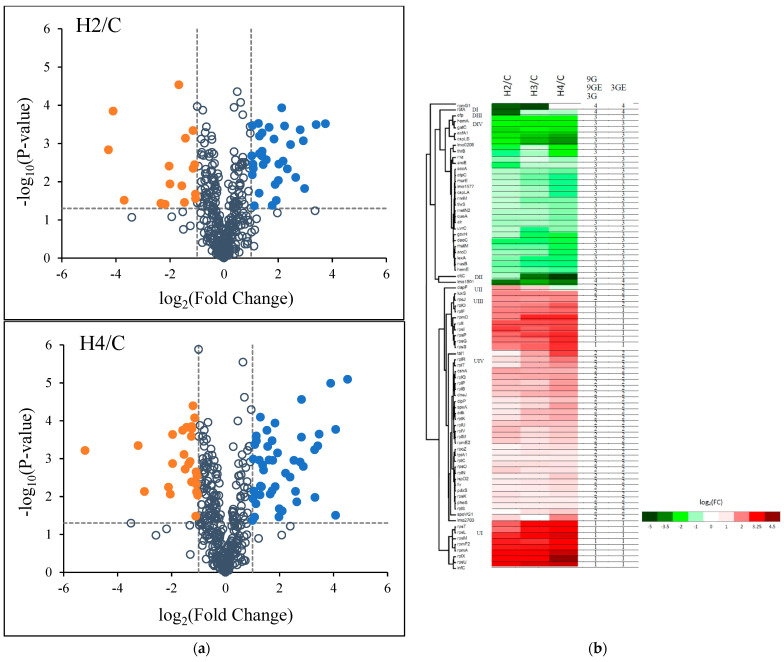
(**a**) Volcano plot of changes in the levels of identified *L. monocytogenes* proteins analyzed using label-free quantitative proteomics under the H2 and H4 treatments. Note: C, control group; H2 and H4, HPP group (200 and 400 MPa). The dashed line of y-axis is *p*-value = 0.05. The dashed lines of x-axis on the left and right side, respectively, are fold change = −2 and 2. Blue dots were upregulated DEPs; orange dots were downregulated DEPs; the others were not DEPs. (**b**) Hierarchical clustering heatmap of differentially expressed proteins (DEPs) of the H2 and H4 treatments. The rows represent individual proteins with gene name on the right of each corresponding row. Red bars indicate upregulated proteins and green bars indicate downregulated proteins. The k-means clustering algorithm identified the 9G, 9GE, 3G and 3GE groups.

**Figure 3 biology-11-01152-f003:**
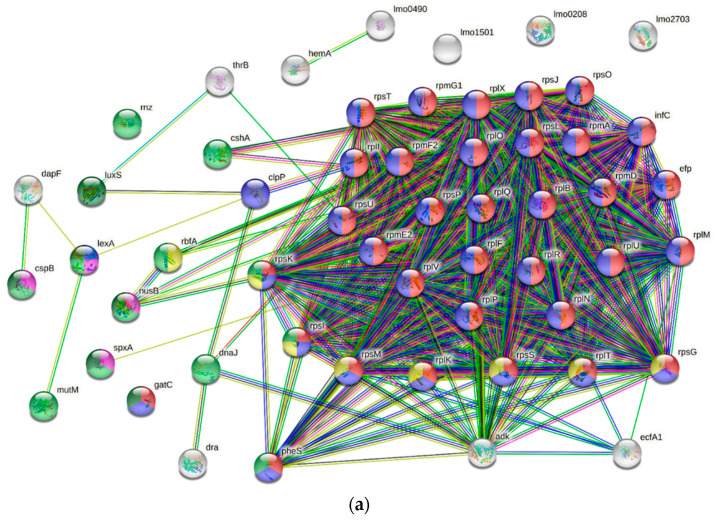
Protein–protein interaction of DEPs in *L. monocytogenes* under the H2 and H4 treatments. Functional categorization was conducted based on gene ontology (GO) level in STRING Protein–Protein Interaction Networks (v. 11.0) (**a**,**b**).

**Figure 4 biology-11-01152-f004:**
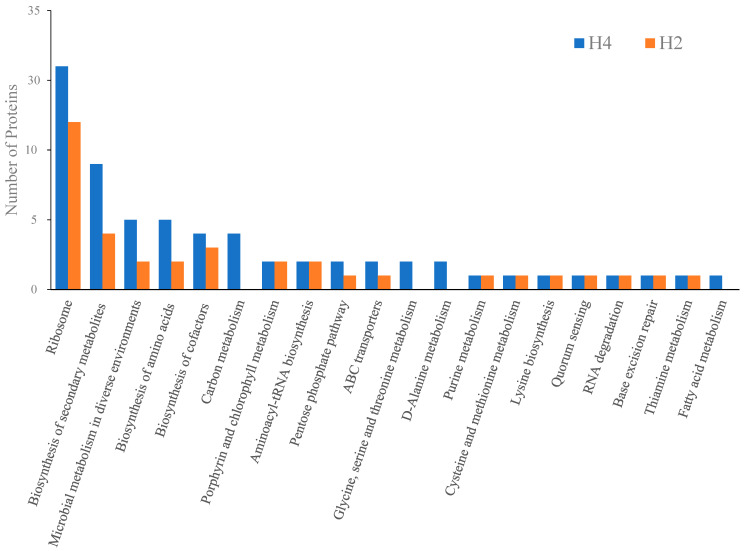
The top 20 pathways using KEGG pathway analysis of differentially expressed proteins (DEPs) in *L. monocytogenes* under the H2 and H4 treatments.

**Table 1 biology-11-01152-t001:** KEGG pathway analysis of the DEPs of *L. monocytogenes* under HPP at 200 and 400 MPa.

Functional Characterization		Upregulation	Downregulation
Metabolism (M)			
Global and overview maps	01100 Metabolic pathways	tal1^2^, adk, tpiA1^2^, pdxS^2^, luxS, tarI^2^, dapF^1^	thrB, hemE, hemA, deoC, Alr^2^, murE^2^, accA^2^, gatC, atpC^2^, gcvH^2^, aroD^2^, dltC^2^
	01110 Biosynthesis of secondary metabolites	adk, tal1^2^, tpiA1^2^, dapF^1^	thrB, hemE, accA^2^, gcvH^2^, hemA, aroD^2^
	01120 Microbial metabolism in diverse environments	adk, tal1^2^, tpiA1^2^, dapF^1^	thrB, accA^2^, hemA
	01200 Carbon metabolism	tal1^2^, tpiA1^2^	accA^2^, gcvH^2^
	01230 Biosynthesis of amino acids	tal1^2^, tpiA1^2^, luxS^2^, dapF^1^	thrB, aroD^2^
Carbohydrate metabolism	00010 Glycolysis/Gluconeogenesis	tpiA1^2^	
	00030 Pentose phosphate pathway	tal1^2^	deoC
	00040 Pentose and glucuronate interconversions	tarI^2^	
	00051 Fructose and mannose metabolism	tpiA1^2^	
	00620 Pyruvate metabolism		accA^2^
	00630 Glyoxylate and dicarboxylate metabolism		gcvH^2^
	00640 Propanoate metabolism		accA^2^
	00562 Inositol phosphate metabolism	tpiA1^2^	
Energy metabolism	00190 Oxidative phosphorylation		atpC2
	ko03029 Mitochondrial biogenesis	infC, rpmF^2^, rpsP, dnaJ	gatC
	00195 Photosynthesis		atpC^2^
	00710 Carbon fixation in photosynthetic organisms	tpiA1^2^	
	00720 Carbon fixation pathways in prokaryotes		accA^2^
Lipid metabolism	00061 Fatty acid biosynthesis		accA^2^
Nucleotide metabolism	00230 Purine metabolism	adk, rpoZ2	
Amino acid metabolism	00260 Glycine, serine and threonine metabolism		thrB, gcvH^2^
	00270 Cysteine and methionine metabolism	luxS	
	00300 Lysine biosynthesis	dapF^1^	murE^2^
	00400 Phenylalanine, tyrosine and tryptophan biosynthesis		aroD^2^
	ko01002 Peptidases and inhibitors	clpP	lexA
Glycan biosynthesis	00473 D-alanine metabolism		Alr^2^, dltC^2^
	00550 ko01011 Peptidoglycan biosynthesis and degradation proteins		Alr^2^, murE^2^, dltC^2^
Metabolism of cofactors and vitamins	00730 Thiamine metabolism	adk	
	00750 Vitamin B6 metabolism	pdxS^2^	
	00860 Porphyrin and chlorophyll metabolism		hemE, hemA
**Genetic Information Processing (GIP)**			
Transcription	03020 RNA polymerase	rpoZ^2^	
	ko03021 Transcription machinery	rpoZ^2^	nusB
	Transcription regulation: stress response		cspLA^2^, cspLB
	ko03019 Messenger RNA biogenesis	cshA	
Translation	03010 Ribosome	rpsE, rpsG, rpsI, rpsJ rpsK, rpsL, rpsM, rpsO, rpsP, rpsS, rpsT, rpsU, rplB, rplC^2^, rplF, rplI, rplK, rplM, rplN, rplO, rplP, rplQ, rplR, rplS^2^, rplT, rplU, rplV, rplX, rpmA, rpmD, rpmE2, rpmF2	rpmG1^1^
	00970 Aminoacyl-tRNA biosynthesis	pheS^1^	thrS^2^, gatC
	ko03009 Ribosome biogenesis	cshA	rimM^2^, nusB, rbfA^1^
	ko03016 Transfer RNA biogenesis	pheS^1^	thrS^2^, queA^2^, gatC, rnz^1^
	ko03012 Translation factors	infC, frr^2^	efp
Folding, sorting and degradation	03013 RNA transport 03018 RNA degradation	cshA	rnz^1^
	ko03110 Chaperones and folding catalysts	dnaJ	
Replication and repair	03410 Base excision repair		mutM
	03420 Nucleotide excision repair		uvrC2
	ko03400 DNA repair and recombination proteins	rpoZ^2^	lexA, uvrC^2^, mutM
**Environmental Information Processing (EIP)**			
Membrane transport	02010 ABC transporters		metN2^2^, ecfA1
Signal transduction	02020 Two-component system		dltC^2^
**Cellular Processes (CP)**			
Cell growth and death	04112 Cell cycle—Caulobacter	clpP	
Cellular community—prokaryotes	02024 Quorum sensing	luxS	
	05111 Biofilm formation—*Vibrio cholerae* 02026 Biofilm formation—*Escherichia coli*	luxS	
Regulation of cell septum	ko04812 Cytoskeleton proteins	spoVG1^2^	
Exosome	ko04147 Regulation of Exosome (2)	adk, tpiA1^2^	
**Organismal Systems (OS)**			
Aging			
	04212 Longevity regulating pathway—worm	clpP	
**Human Diseases (HD)**			
Infectious disease: bacterial	05150 *Staphylococcus aureus* infection		dltC^2^
Drug resistance: antimicrobial	01502 Vancomycin resistance		Alr^2^
	01503 Cationic antimicrobial peptide (CAMP) resistance		dltC^2^

^1^: DEP only detected in H2/C. ^2^: DEP only detected in H4/C.

## Data Availability

Not applicable.

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
