# Peer review of "Differential Proteomic Analysis of Listeria monocytogenes during High-Pressure Processing"

_biology, 2022, doi:10.3390/biology11081152_

Round 1

Reviewer 1 Report

The introduction and MM section are substantial wìwell written, but I suggest a deeper revision of Results and Discussion. Authors cited H3 (a new different treatment and alsio discussion is based on it BUT this is not interesting for study,or yes? And if it is yes, it not present in MM and not in all results.

It seems that H3 is cited as a boring way to not rebuilt clustering or other analysis which were probably just prepared.

Author Response

Reviewer 1:

Thank you for careful revision on our manuscript. We separated the section [3. Results and Discussion] into [3. Results] and [4. Discussion] according to reviewer’s suggestion. A deeper revision has been made in the revised edition. The H3 in this study was only applied in the [2.6. Machine Learning of Grouping Proteins] because a big data is essential for machine learning, the co-author (Dr. Ku) insisted. The aim of this study was to compare proteomic changes of L. monocytogenes under HPP at 200 and 400 MPa, which stand for sublethal to lethal injuries. HPP showed similar bactericidal efficacy at 200-300 MPa and reduced <1 and <2 log CFU/mL for 200 and 300 MPa, respectively, but completely inactivated (initial bacterial load was 7.35 log CFU/mL) at 400 MPa (Table S1). Therefore, we selected the 200 and 400 MPa due to a bigger difference. Duru et al. (2021) agreed that HPP at 200 MPa was ineffective to inactivate L. monocytogenes RO15 and ScottA strains, but 400 MPa significantly reduced 5.78-7.04 log CFU/mL for RO15 and ScottA, respectively [12]. On the other hand, H3 was previous published while comparing 20 ppm of slightly acidic electrolyzed water, H3 and their combination for inactivation of L. monocytogenes by our team. But we did not perform machine learning and did not use any data of H2 and H4 in our previous report. Hence, the whole manuscript was emphasized on the comparison between H2 and H4, except for machine learning. Thank you for understanding.

Reviewer 2 Report

In the submitted manuscript, Chen et.al. used label-free quantitative proteomics analysis to investigate the change of Listeria proteome during high-pressure processing. The authors identified pathways in listeria that response to high-pressure stress. Overall, the study is well conducted.

I have some minor suggestions for the authors to consider:

Revise the sentence in line “50-51”.  

In the submitted manuscript, Chen et.al. used label-free quantitative proteomics approach to investigate the change of Listeria proteome during high-pressure processing. The authors identified proteins and pathways in listeria that response to high-pressure stress. Overall, the study is well conducted. However, some text editing is needed to improve the quality the manuscript. Current version of manuscript is not easy for readers to understand.  
Here are some additional suggestions for the authors to consider:
1. Change the title from “Differential proteomics expression of” to either “Differential protein expression of” or “Differential proteomic analysis of ”.
2. Revise sentence in line 47. It is hard to understand the following sentence: “which can be employed to detect given cells under specific conditions.” 
3. Revise the sentence in line “50-51”. 
4. Line 67 “The L. monocytogenes culture was enrich with…”. The word “enrich” is usually used under very specific terms, for instance, enrich listeria from a complex mixture of microbes. I would suggest the authors change to “The L. monocytogenes was cultured in”.
5. Line 72. Delete “composition”.
6. Line 76: Explain ca.
7. Change “Labelled C” to “Labelled as C”, change “labelled H2” to labelled as H2”.
8. Revise the sentence in line 107 “Three biological replications were performed for the control (C), H2, H3, and H4 groups. ”
9. Revise the sentence in 110-111: “The treatment was considered to have a significant influence on protein expression if a 2-fold increase or decrease in protein expression level was observed (P < 0.05) ”.
10. Revise the sentences in line 177-180. Current version is hard to understand.
11. In line 177 to line 203, the authors listed a colossal amount of protein names. Since these proteins could be found in tables, simply putting these protein names in main text is not necessary.

Author Response

Reviewer 2:

Thank you for careful revision on our manuscript. We separated the section [3. Results and Discussion] into [3. Results] and [4. Discussion] according to reviewer’s suggestion. A deeper revision has been made in the revised edition.

  1. We have changed the title from “Differential proteomics expression of” to “Differential proteomic analysis of ”.
    2. Revised sentence in line 47 to line 60.
    3. Revised the sentence in line “50-51” to 81-82. We have provided more detailed information on HPP induced gene set enrichment analysis in L. monnocytogenes from line 63-80.
    4. Line 67 changed to “The L. monocytogenes was cultured in” (line 100)
    5. Line 72. Delete “composition” (line 105)
    6. Line 76: Explain ca. We revised to the correct level (line 109)
    7. Changed “Labelled C” to “Labelled as C”, change “labelled H2” to labelled as H2”.(line 118)
    8. Revised the sentence in line 107 to line 140.
    9. Revised the sentence in 110-111 to line 143-146.
    10. Revise the sentences in line 177-180. We have reduced the redundancy statements.
    11. In line 177 to line 203, We have reduced the redundancy statements, and separated Results and Discussion.

Reviewer 3 Report

General comment on language: The manuscript needs light to medium editing for spelling (be consistent when using hyphenated words, use of plural vs singular), punctuation (, and ;), article usage, correct use of verb tense, verb/subject agreement, and make sure each sentence has a verb, and check “ing” vs “ed” word ending. Overall need to check basic spelling and grammar. In addition, sentence structure should be revised so that sentences are not as lengthy, which leads to difficult reading and confusion. Remember to prefer using a verb rather than its noun form.

 The objective of the present study was to evaluate the efficacy and mechanism of the HPP at 200 and 400 MPa in inactivating L. monocytogenes (from sub-lethal to lethal injuries) by label-free quantitative proteomics.

Pros: Interesting study that shows evidence with respect to the efficacy of HPP in inactivating L. monocytogenes using proteomics.

Cons: Some aspects should be included in M&M, such as the experimental design (which the authors mention in the Results section), and eliminate some parts of the text that are not justified. The main problem that undermines the manuscript is how the results and discussion are combined. The way these two sections are presented is confusing, repetitive, and provides poor readability to the text because the written results are also shown in the tables. Therefore, the two sections should be separated or the results section should be presented with a discussion that supports the findings.

L19: define KEGG

L23: define Clp

L24: define Spx and PrfA

L24: revise regulon?

L25-26. The authors comment on the contribution of this study to listeriosis; however, two of the most important factors associated with the disease are Hly and InlA proteins, which are not studied or mentioned.

L32: Which bacteria?

L32-35: Your statements need a reference.

L54-57: This is part of the methodology and should be eliminated here.

Introduction: You should include a paragraph on several studies that have been conducted on Listeria monocytogenes by applying HHP. In addition, the effects on the physiology and genetic material of L. monocytogenes should be included.

Materials and methods: Revise the similarity of the methodology with the manuscript by the same group of authors:  Combined impact of high-pressure processing and slightly acidic electrolysed water on Listeria monocytogenes proteomes. Doi: https://doi.org/10.1080/87559129.2015.1094816

For a better understanding of the results, include the experimental design that was used and describe the subsequent stages.

Rewrite subsection 2.6.1. Use the methodology by the same group of author.  Doi: https://doi.org/10.1080/87559129.2015.1094816

L129-155: This could be written as supplementary information.

L160-164: For better understanding by the reader, use a figure that describes the effect on survival and bacteria growth.

L70-171: The authors express the development of an experiemental design with replicates, but this is not included in the M&M.

Results and discussion

Line 169-360: Only results are indicated with no discussion.

L194-211: This information is very confusing and is repeated in Table 1. I suggest you separate the results and discussion sections to provide better understanding.

Why do the authors comment in the abstract about the stimulation of the PrfA protein and Spx expression (L24) by HPP and these results do not appear in Tables 1 and 2 in the results?

L383-451: Only results are mentioned and no discussion.3

L476 and 484: Why is this text in bold face?

Why did the authors not analyze the internalins, cassettes (InlAbC), and Hly protein?

Revise to see if Table 2 is pertinent or can be included in supplementary material.

Join Figures 2a and 2b. The location of Figure 2b in the manuscript does not make sense.

Author Response

Reviewer 3:

Thank you for careful revision on our manuscript. We separated the section [3. Results and Discussion] into [3. Results] and [4. Discussion] according to reviewer’s suggestion. A deeper revision has been made in the revised edition. We have re-check the whole manuscript more carefully for consistency and less-redundancy.

L19:L31 KEGG was defined

L23:L35 Clp was defined

L24:L35-36 Spx and PrfA were defined

L24:L37 revise regulon? It was deleted.

L25-26. The authors comment on the contribution of this study to listeriosis; however, two of the most important factors associated with the disease are Hly and InlA proteins, which are not studied or mentioned.

Ans: The proteomic data provided us a total 380 proteins but we cannot find Hly and InlA proteins. Hence, we are sorry that we just discussed a few in our manuscript.

L32:L45 Which bacteria? L. monocytogenes

L32-35:L48 Your statements need a reference. We put the [2] reference.

L54-57:L150-155 This is part of the methodology and should be eliminated here. We combined into ‘’2.5. Data analysis and protein network construction’’.

Introduction: You should include a paragraph on several studies that have been conducted on Listeria monocytogenes by applying HHP. In addition, the effects on the physiology and genetic material of L. monocytogenes should be included.

Ans: We have added into line 63-85.

Materials and methods: Revise the similarity of the methodology with the manuscript by the same group of authors:  Combined impact of high-pressure processing and slightly acidic electrolysed water on Listeria monocytogenes proteomes. Doi: https://doi.org/10.1080/87559129.2015.1094816

Ans: We have rewritten and shorten the sentences in the ‘’Materials and Methods’’ and ‘’Discussion’’.

For a better understanding of the results, include the experimental design that was used and describe the subsequent stages.

Rewrite subsection 2.6.1. Use the methodology by the same group of author.  Doi: https://doi.org/10.1080/87559129.2015.1094816

Ans: We did not find the similarity between these two manuscripts.

L129-155: This could be written as supplementary information.

Ans: We decide to let the algorithm in the main text and put the COGs into supplementary Table S4.

L160-164: For better understanding by the reader, use a figure that describes the effect on survival and bacteria growth.

Ans: We used 100-400 MPa and just 4 points. Hence, we think it is concise.

L170-171: The authors express the development of an experimental design with replicates, but this is not included in the.

Ans: We have moved to M&M (line 152-156)

Results and discussion

Line 169-360: Only results are indicated with no discussion.

Ans: we have separated Results and Discussion to make it clear and concise.

L194-211: This information is very confusing and is repeated in Table 1. I suggest you separate the results and discussion sections to provide better understanding.

Ans: Thank you for the wonderful suggestion, and we have separated Results and Discussion to make it clear and concise.

Why do the authors comment in the abstract about the stimulation of the PrfA protein and Spx expression (L24) by HPP and these results do not appear in Tables 1 and 2 in the results?

Ans: We actually did not find the PrfA in our proteomic data but Spx and Clp were DEPs shown in the Supplementary Table S3-S4. So that we tried to discuss the roles of PrfA and sigB indirectly from Spx, Clp proteases and GSH.

L383-451: Only results are mentioned and no discussion.3

Ans: We put into Discussion.

L476 and 484: Why is this text in bold face? We revised it.

Why did the authors not analyze the internalins, cassettes (InlAbC), and Hly protein?

Ans: The proteomic data provided us a total 380 proteins but we cannot find Hly and InlA proteins. Hence, we are sorry that we just discussed a few in our manuscript.

Revise to see if Table 2 is pertinent or can be included in supplementary material.

Ans: We put the COGs into supplementary Table S4.

Join Figures 2a and 2b. The location of Figure 2b in the manuscript does not make sense.

Ans: The layout was totally missed because of the Word software. And we tried to replace by the image.

Reviewer 4 Report

This paper studied Differential proteomics expression of Listeria monocytogenes during high-pressure processing, 

  1. However, this research on proteomic differential expression of Listeria is now common. What is your innovation? 
  2. The section of Results and Discussion is almost from the test report and is not sufficiently discussed

Author Response

Reviewer 4:

Thank you for careful revision on our manuscript. We separated the section [3. Results and Discussion] into [3. Results] and [4. Discussion] according to reviewer’s suggestion. A deeper revision has been made in the revised edition. The quantitative proteomics becomes popular and user-friendly these years. From line 63-87 in the [Introduction], so far as we know that no studies aim to use quantitative proteomic technology to explore HPP induced L. monocytogenes sublethal to lethal injuries. We think this work might be potential to readers for better understanding of HPP inactivate L. monocytogenes. Thank you very much for careful revision and suggestions.

Round 2

Reviewer 3 Report

Line 159: reference 119 or 19?

Reviewer 4 Report

accept